# Sociodemographic and clinical risk factors for paediatric typical haemolytic uraemic syndrome: retrospective cohort study

Natalie Adams ![ORCID],[1,2] Lisa Byrne,[2] Tanith Rose,[1,3] Bob Adak,[1] Claire Jenkins,[1,2] Andre Charlett,[2] Mara Violato,[1,4] Sarah O'Brien,[1,3] Margaret Whitehead,[1,3] Benjamin Barr,[1,3] David Taylor-Robinson,[1,3] Jeremy Hawker[1,5]

► Additional material is published online only. To view, please visit the journal online (http://dx.doi.org/10.1136/bmjpo-2019-000465)

[1]Health Protection Research Unit in Gastrointestinal Infections, National Institute for Health Research, Liverpool, UK
[2]National Infection Service, Public Health England, London, UK
[3]Department of Public Health and Policy, University of Liverpool, Liverpool, UK
[4]Health Economics Research Centre, University of Oxford, Oxford, UK
[5]National Infection Service, Public Health England, Birmingham, UK

**Correspondence to**
Dr Benjamin Barr; b.barr@liverpool.ac.uk

## ABSTRACT

**Objectives** Haemolytic uraemic syndrome (HUS) following Shiga toxin-producing *Escherichia coli* (STEC) infection is the the most common cause of acute renal failure among children in the UK. This study explored differential progression from STEC to HUS by social, demographic and clinical risk factors.

**Methods** We undertook a retrospective cohort study linking two datasets. We extracted data on paediatric STEC and HUS cases identified in the Public Health England National Enhanced Surveillance System for STEC and British Paediatric Surveillance Unit HUS surveillance from 1 October 2011 to 31 October 2014. Using logistic regression, we estimated the odds of HUS progression by risk factors.

**Results** 1059 paediatric STEC cases were included in the study, of which 207 (19.55%, 95% CI 17% to 22%) developed HUS. In the fully adjusted model, the odds of progression to HUS were highest in those aged 1–4 years (OR 4.93, 95% CI 2.30 to 10.56, compared with 10–15 years), were infected with an Shiga toxin (*stx*) 2-only strain (OR 5.92, 95% CI 2.49 to 14.10), were prescribed antibiotics (OR 8.46, 95% CI 4.71 to 15.18) and had bloody diarrhoea (OR 3.56, 95% CI 2.04 to 6.24) or vomiting (OR 4.47, 95% CI 2.62 to 7.63), but there was no association with progression to HUS by socioeconomic circumstances or rurality.

**Conclusion** Combining data from an active clinical surveillance system for HUS with the national enhanced STEC surveillance system suggests that 20% of diagnosed paediatric STEC infections in England resulted in HUS. No relationship was found with socioeconomic status or rurality of cases, but differences were demonstrated by age, *stx* type and presenting symptoms.

## INTRODUCTION

Haemolytic uraemic syndrome (HUS) is a rare but serious complication of infection with Shiga toxin-producing *Escherichia coli* (STEC), affecting the blood, kidneys and, in the most severe cases, the central nervous system. Children and the elderly are considered to be the most susceptible age groups, and HUS is recognised as the most common

### What is known about the subject?

► Haemolytic uraemic syndrome (HUS) is recognised as the most common cause of acute renal failure among children in the UK.
► It is estimated that progression to HUS following Shiga toxin-producing *Escherichia coli* (STEC) infection could be as high as 15% in young children.
► Several studies have suggested that development of HUS varies by demographic characteristics; however, few have documented progression to HUS by demographic characteristics.

### What this study adds?

► A fifth of paediatric STEC cases developed the serious complication of HUS in England.
► This figure is higher than previously reported in England and varied by demographic and clinical factors.
► Socioeconomic factors did not influence progression to HUS.

cause of acute renal failure among children in the UK.[1] Strains of STEC encoding *stx2* toxin genes are more often associated with HUS than other strains.[1–6] STEC serogroup O157 is the most frequently reported strain causing illness in England. Transmission to humans occurs through consumption of contaminated food or water, exposure to a contaminated environment involving direct or indirect contact with animals or their faeces and person-to-person spread.

It is estimated that progression to HUS following STEC infection could be as high as 15% in young children.[2 7] Several studies have suggested that development of HUS varies by some demographic characteristics; higher incidence of HUS has been documented in children (particularly those aged 1–4 years),

girls (particularly those aged over 10 years) and in those of white ethnicity[2 6 8–13]; however, few have documented progression to HUS by other demographic characteristics such as deprivation, foreign travel, rurality or region. There is evidence to suggest that those who are disadvantaged have a lower risk of STEC infection[14–16] and potentially a lower risk of progression to HUS outside of England[16 17]; however, no studies have looked at the relationship between socioeconomic status (SES), STEC and HUS in England. This study aimed to investigate the relationship between demographic factors, STEC infection and subsequent development of HUS in a well-characterised paediatric population in England with high case ascertainment.

## METHODS
### Data, setting and source

We undertook a retrospective cohort study linking two data sources: the Public Health England (PHE) National Enhanced Surveillance System for STEC (NESSS) and the British Paediatric Surveillance Unit (BPSU) HUS Study in conjunction with PHE. The linkage of two robust datasets, both of which can record HUS status, ensures high ascertainment of HUS cases. First, we extracted data on STEC cases aged 0–15 years (inclusive) identified in NESSS during the period of the BPSU HUS Study (1 October 2011 to 31 October 2014). All laboratory-confirmed STEC cases in England are reported by National Health Service (NHS) laboratories to PHE staff who collect standardised data through an Enhanced Surveillance Questionnaire (ESQ) as part of their public health response: this standardised dataset is collated centrally in NESSS for further validation and analysis. The ESQ collects detailed information on patient demographics, symptoms, food and water exposures, and UK and non-UK travels during the exposure period (the week prior to illness onset). When a presumptive STEC is identified at the local laboratory or a case of HUS is identified, specimens are sent to the PHE Gastrointestinal Bacteria Reference Unit for testing, and patient ESQs are linked to microbiological results. Due to the timing of the ESQ administration in NESSS (which is designed to inform the acute public health response), this system can underascertain HUS as this can develop after completion of the questionnaire. This surveillance system is described in detail elsewhere.[2]

Second, we extracted clinical data on paediatric (<16 years) HUS cases, collected by the BPSU HUS Study, an active surveillance system requiring regular returns from clinicians. Within this study, data were captured using a standardised questionnaire administered to paediatricians collecting information on case demography, treatment history, microbiological investigations, clinical parameters of illness, clinical management of illness and status of the case at the time of data capture. Cases in the BPSU dataset were linked on the NHS number, which was available for all cases, to those in the NESSS dataset to create a retrospective cohort. Online

supplementary figure 1 provides details of the selection of study participants.

For statistical analysis, cases for whom no microbiological information was available (n=4) and cases identified via serological testing only (n=66) were excluded in order to assess the role of the Shiga toxin (*stx*) subtype. Ethnic groups, collected in five categories (white, Asian/Asian British, black/black British, mixed and Chinese), are not well-completed in NESSS, and therefore responses were recoded as white or non-white for analysis. The considerable missing data for the ethnicity variable (19.1%) has led us to use the crude dichotomy of white/non-white in this analysis. Multiple imputation using chained equations was used to impute values where ethnicity (white/non-white) was missing. There will clearly be some loss of information from doing this, and this precludes investigating risk differences between the non-white ethnic groups. This may also slightly affect the confounding that exists between ethnicity and SES. Fifty imputed datasets were generated. The distribution of ethnicity by age, sex and region was assessed to check the missing at random assumption. There was no difference in missing ethnicity by sex; however, there were some differences by age group (57.3% of cases with missing ethnicity were in the 1–4 years age group, n=114/199) and region (31.2% of cases with missing ethnicity were in London, n=62/199); these were not regarded as problematic, however as, given the observed data for other variables, the missing data were considered independent.

### Patient and public involvement

Patients were not directly involved in the design of this study.

### Outcome and exposures

The outcome of interest was HUS, determined by the case meeting the BPSU clinical criteria (see online supplementary table 1)[18] or completion of the HUS field in the ESQ. Covariates in the analysis were age group (<1, 1–4, 5–9 and 10–15 years); sex (male/female); ethnicity (white/non-white); travel (yes/no); rurality (rural/urban); microbiology (*stx*); antibiotic use (yes/no); clinical symptoms (diarrhoea, bloody diarrhoea, nausea, vomiting, abdominal pain and fever) and region of residence. The s*tx* type, the primary STEC virulence factor, was used as the main microbiological variable.[19] Where symptoms, travel status and healthcare contact variables were blank or unknown, these were recoded as a negative response. As a proxy for childhood socioeconomic circumstances (SECs), we used a small-area deprivation measure, the Index of Multiple Deprivation 2010 (IMD),[20] assigned to each case based on their postcode and divided into population-level quintiles, with the first quintile representing the least deprived and the fifth quintile representing the most deprived.

## Analysis strategy

Comparisons of proportions were tested using the $\chi^2$ test. We explored univariate relationships between progression to HUS and the covariates of interest before fitting a multivariate logistic regression model. All variables were retained in this model in order to control for any potential confounding. Interaction terms between variables (IMD, ethnicity, age and sex) were tested to investigate whether the strength of any relationship was moderated by the inclusion of another variable. Analyses were conducted in STATA V.13.1.

## Robustness tests

We performed multiple sensitivity analyses to test the validity of the main analysis by (1) excluding cases that were likely to have a travel-acquired STEC infection (date of onset is within one exposure period, 7 days, of having returned from outside of the UK) and (2) separately excluding cases with unknown ethnicity to determine whether there were differences in progression to HUS by SECs for children who travelled abroad during their incubation period compared with those who did not or those with ethnicity recorded and those without, respectively.

## RESULTS

### Descriptive analysis

Of 1059 paediatric STEC cases included in the study, 207 (19.55%, 95% CI 17.27% to 22.04%) developed HUS. Progression to HUS varied by age and gender (table 1); the highest was observed in girls aged 1–4 years (26.0%). A higher proportion of progression to HUS was observed in girls aged 10–15 years compared with boys of the same age (19.3%, 95% CI 12.3% to 27.9% vs 7.1%, 95% CI 2.9% to 14.2%; p=0.01), and among girls aged less than 1 year compared with boys of the same age, although this was not significant (14.3%, 95% CI 4.0% to 32.7% vs 4.8%, 95% CI 0.6% to 16.2%; p=0.16). Although progression to HUS was higher in the least disadvantaged quintile (47/245; 19.2%, 95% CI 14.4% to 24.7%) compared with the most disadvantaged quintile (29/189; 15.3%, 95% CI 10.5% to 21.3%), this difference was not statistically significant (p=0.29). The highest proportion progressing to HUS was in quintile 3 (53/219; 24.2%, 95% CI 18.7% to 30.4%), and there was no clear pattern across the five quintiles (p=0.07; quintile 2: 35/221, 15.8%, 95% CI 11.3% to 21.3%; quintile 4: 43/185, 23.2%, 95% CI 17.4% to 30%).

### Multivariable analysis

In the fully adjusted model (table 2), there were significantly lower odds of HUS among children aged <1, 5–9 and 10–15 years compared with those aged 1–4 years and significantly higher odds of HUS among those infected with *stx2*-only strains, those prescribed antibiotics and those who had experienced bloody diarrhoea or vomiting . The most disadvantaged children had lower odds of

**Table 1** Characteristics of cohort participants by HUS status (N=1059)

| | | No HUS n (%) | HUS n (%) |
|---|---|---|---|
| Total | | 852 (80.5) | 207 (19.6) |
| Age group (years) | <1 | 64 (91.4) | 6 (8.6) |
| | 1–4 | 370 (76.1) | 116 (23.9) |
| | 5–9 | 239 (80.7) | 57 (19.3) |
| | 10–15 | 179 (86.5) | 28 (13.5) |
| Sex | Female | 400 (77.5) | 116 (22.5) |
| | Male | 452 (83.2) | 91 (16.8) |
| Age and sex | Female <1 | 24 (85.7) | 4 (14.3) |
| | Female 1–4 | 171 (74.0) | 60 (26.0) |
| | Female 5–9 | 117 (79.1) | 31 (20.9) |
| | Female 10–15 | 88 (80.7) | 21 (19.3) |
| | Male <1 | 40 (95.2) | 2 (4.8) |
| | Male 1–4 | 199 (78.0) | 56 (22.0) |
| | Male 5–9 | 122 (82.4) | 26 (17.6) |
| | Male 10–15 | 91 (92.9) | 7 (7.1) |
| Ethnicity | White | 552 (80.5) | 134 (19.5) |
| | Non-white | 138 (88.5) | 18 (11.5) |
| | Unknown | 162 (74.7) | 55 (23.4) |
| IMD quintile | 1 (least disadvantaged) | 198 (80.8) | 47 (19.2) |
| | 2 | 186 (84.2) | 35 (15.8) |
| | 3 | 166 (75.8) | 53 (24.2) |
| | 4 | 142 (76.8) | 43 (23.2) |
| | 5 (most disadvantaged) | 160 (84.7) | 29 (15.3) |
| Travel | Yes | 128 (85.3) | 22 (14.7) |
| | No | 724 (79.7) | 185 (20.4) |
| Rurality | Rural | 230 (80.4) | 56 (19.6) |
| | Urban | 622 (80.5) | 151 (19.5) |
| Region | East Midlands | 65 (81.3) | 15 (18.8) |
| | East of England | 57 (80.3) | 14 (19.7) |
| | London | 93 (81.6) | 21 (18.4) |
| | North East | 64 (77.1) | 19 (22.9) |
| | North West | 153 (77.7) | 44 (22.3) |
| | South East | 92 (78.6) | 25 (21.4) |
| | South West | 101 (75.9) | 32 (24.1) |
| | West Midlands | 96 (84.2) | 18 (15.8) |
| | Yorkshire and Humber | 131 (87.3) | 19 (12.7) |
| *stx* | *stx1* | 17 (94.4) | 1 (5.6) |
| | *stx2* | 609 (81.7) | 136 (18.3) |
| | *stx1+2* | 219 (96.9) | 7 (3.1) |
| | Serology | 7 (10.6) | 59 (89.4) |
| | Unknown | 0 (0.0) | 4 (100.0) |

Continued

**Table 1** Continued

| | | No HUS | HUS |
|---|---|---|---|
| | | n (%) | n (%) |
| Symptoms | Diarrhoea | 803 (80.3) | 197 (19.7) |
| | Bloody diarrhoea | 432 (74.0) | 152 (26.0) |
| | Nausea | 278 (75.8) | 89 (24.3) |
| | Vomiting | 330 (66.1) | 169 (33.9) |
| | Abdominal pain | 574 (78.2) | 160 (21.8) |
| | Fever | 273 (76.7) | 83 (23.3) |
| Healthcare contact | Antibiotics | 53 (40.8) | 77 (59.2) |
| | NHS Direct | 67 (72.0) | 26 (28.0) |
| | GP | 570 (83.7) | 111 (16.3) |
| | A&E | 186 (66.9) | 92 (33.1) |
| | Other healthcare contact | 98 (74.8) | 33 (25.2) |
| | Hospital | 223 (52.4) | 203 (47.6) |

NHS Direct is the NHS telephone advice line, now NHS 111.
A&E, accident and emergency; GP, general practitioner; HUS, haemolytic uraemic syndrome; IMD, Index of Multiple Deprivation 2010; NHS, National Health Service; *stx*, Shiga toxin.

progression to HUS compared with the least disadvantaged children (OR 0.57, 95% CI 0.25 to 1.31), but the difference was not significant. There was no statistically significant difference in risk by rurality (OR 0.88, 95% CI 0.52 to 1.48) or by region (table 2). There were no significant interactions identified (data not shown).

The sensitivity analyses conducted to assess the robustness of the findings did not alter the overall conclusions of this research (online supplementary tables 2 and 3).

## DISCUSSION

In a novel linkage and analysis of two datasets with high case ascertainment to explore the role of demographic and socioeconomic factors in the development of HUS following STEC infection, we found progression from STEC infection to HUS to be 20% in this paediatric cohort in England. Odds of HUS progression varied by age, *stx* type, antibiotic exposure and clinical presentation, with children aged 1–4 years infected with *stx2* only, with reported antibiotic exposure and presenting with bloody diarrhoea or vomiting at highest risk. Few studies have explored the social patterning of risk factors for STEC[21] or the sociodemographic risk factors associated with progression to HUS, and no such studies have been undertaken in England. We found no relationship between progression to HUS and SES in children in this study.

Our study has several strengths. This study captures the progression of HUS in a well-characterised paediatric STEC population. To the best of our knowledge, as confirmed by a prior review of the literature and discussion with national experts, this is the first study to

combine a prospective active surveillance system and a multisource national surveillance system to study the risk factors for HUS and, as such, is likely to have better case ascertainment of HUS than previous studies and is related to good STEC denominators. Furthermore, this study makes use of one of the largest cohorts of HUS cases. The results of this study are likely to be generalisable to other high-income countries with a similar pattern of STEC infection. Despite this, there are some limitations. It is possible that there is residual confounding that could not be controlled for, such as intrinsic childhood characteristics which may increase differential vulnerability or susceptibility by SEC, such as genetic predisposition, comorbidities, and clinical or treatment characteristics. Further, as an area-level measure of SEC was used, it is possible that it may not have been sensitive enough to detect the effect of socioeconomic inequalities, particularly if individual factors rather than area-level factors have more influence over the risk of acquiring more severe strains of STEC with increased risk of progression to HUS. However, person-to-person spread is an important risk factor for GI infections, and, although there is a risk of ecological fallacy, area-level measures have the advantage of including potential environmental factors such as housing and living environment deprivation, which are likely to be important factors in considering individual risk of infection. Excluding individuals with a serological result only from the statistical analysis may introduce a potential bias leading to an underestimate of HUS incidence, which may be important if there are geographical or host factors which are linked to severity of illness, although the number of serology-only diagnoses was small. In England, most diagnosed cases of STEC are of serogroup O157 (95% in our study), and it is possible that our results may be biassed towards the relationship between STEC O157 and progression to HUS, which may differ if other, possibly less pathogenic, serogroups predominate. It is possible, however, that the risk of progression to HUS could be different in populations exposed to STEC organisms with a lower proportion of *stx2*-only producing strains, or with a different age distribution of cases. There were also some missing data in our study, particularly for ethnicity, which we addressed using multiple imputation. The binary ethnicity variable used (white/non-white) was also crude and adopted because of data quality issues in NESSS for this variable. However, a previous study using this data[2] demonstrated differences in risk of STEC between white and non-white ethnic groups (rate ratio 1.43, p<0.001) and so was important to assess in our study, although its inclusion may mask differences in SES. No data were available on whether the children included in our study had underlying or chronic conditions which may be related to their risk of developing HUS. Finally, it was not always possible to determine whether antibiotics had been prescribed during treatment for STEC infection or following a diagnosis of HUS; therefore, the relevant association should be interpreted with caution.

**Table 2** Adjusted and unadjusted regression analysis (n=989)

| Variable | Category | n (%) | Unadjusted OR | (95% CI) | Adjusted* OR | (95% CI) | P value† |
|---|---|---|---|---|---|---|---|
| Age group (years) | <1 | 67 (6.8) | 0.19 | (0.06 to 0.62) | **0.21** | **(0.05 to 0.82)** | **0.03** |
| | 1–4 | 456 (46.1) | 1.0 (reference) | | 1.0 (reference) | | |
| | 5–9 | 274 (27.7) | 0.62 | (0.40 to 0.94) | **0.43** | **(0.25 to 0.74)** | **0.002** |
| | 10–15 | 192 (19.4) | 0.34 | (0.19 to 0.61) | **0.20** | **(0.09 to 0.43)** | **<0.001** |
| Sex | Male | 513 (51.9) | 1.0 (reference) | | 1.0 (reference) | | |
| | Female | 476 (48.1) | 1.37 | (0.96 to 1.96) | 1.38 | (0.88 to 2.14) | 0.16 |
| Ethnicity‡ | White | 797 (80.6) | 1.0 (reference) | | 1.0 (reference) | | |
| | Non-white | 192 (19.4) | 0.39 | (0.18 to 0.81) | **0.28** | **(0.11 to 0.74)** | **0.01** |
| Travel | No | 850 (86.0) | 1.0 (reference) | | 1.0 (reference) | | |
| | Yes | 139 (14.0) | 0.46 | (0.24 to 0.88) | 0.64 | (0.28 to 1.45) | 0.28 |
| Rurality | Urban | 719 (72.7) | 1.0 (reference) | | 1.0 (reference) | | |
| | Rural | 270 (27.3) | 1.21 | (0.82 to 1.77) | 0.88 | (0.52 to 1.48) | 0.63 |
| IMD quintile | 1 (least disadvantaged) | 231 (23.4) | 1.0 (reference) | | 1.0 (reference) | | |
| | 2 | 210 (21.2) | 0.83 | (0.48 to 1.42) | 0.64 | (0.32 to 1.27) | 0.20 |
| | 3 | 204 (20.6) | 1.28 | (0.77 to 2.12) | 1.01 | (0.54 to 1.91) | 0.97 |
| | 4 | 170 (17.2) | 1.10 | (0.64 to 1.90) | 1.10 | (0.54 to 2.26) | 0.79 |
| | 5 (most disadvantaged) | 174 (17.6) | 0.57 | (0.30 to 1.06) | 0.57 | (0.25 to 1.31) | 0.18 |
| Region | East Midlands | 72 (7.3) | 0.62 | (0.24 to 1.59) | 0.59 | (0.18 to 1.92) | 0.39 |
| | East of England | 66 (6.7) | 1.03 | (0.44 to 2.42) | 1.12 | (0.37 to 3.37) | 0.84 |
| | London | 108 (10.9) | 1.0 (reference) | | 1.0 (reference) | | |
| | North East | 76 (7.7) | 1.19 | (0.53 to 2.64) | 0.71 | (0.26 to 1.97) | 0.51 |
| | North West | 185 (18.7) | 1.20 | (0.63 to 2.31) | 1.02 | (0.44 to 2.37) | 0.97 |
| | South East | 107 (10.8) | 1.09 | (0.52 to 2.28) | 1.31 | (0.48 to 3.63) | 0.60 |
| | South West | 127 (12.8) | 1.48 | (0.75 to 2.93) | 1.25 | (0.50 to 3.13) | 0.63 |
| | West Midlands | 104 (10.5) | 0.54 | (0.23 to 1.29) | 0.53 | (0.18 to 1.53) | 0.24 |
| | Yorkshire and Humber | 144 (14.6) | 0.62 | (0.29 to 1.33) | 0.52 | (0.20 to 1.34) | 0.17 |
| *stx* | *stx1+2* | 226 (22.9) | 1.0 (reference) | | 1.0 (reference) | | |
| | *stx1* | 18 (1.8) | 1.84 | (0.21 to 15.84) | 5.53 | (0.53 to 57.42) | 0.15 |
| | *stx2* | 745 (75.3) | 6.99 | (3.22 to 15.17) | **5.92** | **(2.49 to 14.10)** | **<0.001** |
| Antibiotics | No | 887 (89.7) | 1.0 (reference) | | 1.0 (reference) | | |
| | Yes | 102 (10.3) | 8.54 | (5.48 to 13.30) | **8.46** | **(4.71 to 15.18)** | **<0.001** |
| Diarrhoea | No | 49 (5.0) | 1.0 (reference) | | 1.0 (reference) | | |
| | Yes | 940 (95.0) | 8.61 | (1.18 to 62.89) | 4.04 | (0.50 to 32.59) | 0.19 |
| Bloody diarrhoea | No | 440 (44.5) | 1.0 (reference) | | 1.0 (reference) | | |
| | Yes | 549 (55.5) | 4.85 | (3.07 to 8.00) | **3.56** | **(2.04 to 6.24)** | **<0.001** |
| Nausea | No | 653 (66.0) | 1.0 (reference) | | 1.0 (reference) | | |
| | Yes | 336 (34.0) | 1.52 | (1.06 to 2.18) | 1.12 | (0.67 to 1.86) | 0.66 |
| Vomiting | No | 549 (55.5) | 1.0 (reference) | | 1.0 (reference) | | |
| | Yes | 440 (44.5) | 6.05 | (3.95 to 9.26) | **4.47** | **(2.62 to 7.63)** | **<0.001** |
| Abdominal pain | No | 309 (31.2) | 1.0 (reference) | | 1.0 (reference) | | |
| | Yes | 680 (68.8) | 1.49 | (0.99 to 2.25) | 0.82 | (0.46 to 1.46) | 0.50 |
| Fever | No | 657 (66.4) | 1.0 (reference) | | 1.0 (reference) | | |
| | Yes | 332 (33.6) | 1.50 | (1.05 to 2.16) | 1.05 | (0.67 to 1.66) | 0.82 |

Continued

**Table 2** Continued

| Variable | Category | n (%) | Unadjusted OR | (95% CI) | Adjusted* OR | (95% CI) | P value† |
|---|---|---|---|---|---|---|---|

Bold values highlights statistically significant results.
*Adjusted for all other covariates in the model.
†Statistical significance of the relationship between HUS and each variable tested using $\chi^2$ test.
‡Multiply imputed variable.
HUS, haemolytic uraemic syndrome; IMD, Index of Multiple Deprivation 2010; *stx*, Shiga toxin.

The finding of 19.5% (95% CI 17% to 22%) of diagnosed STEC cases progressing to HUS is higher than those of previous studies, which have estimated the proportion of paediatric cases of STEC O157 progressing to HUS to be 15% (95% CI 11% to 19%) in girls aged 1–4 years in England[2] and 15.3% (95% CI 13% to 18%) in children aged <5 years in the USA.[7] Our study uses data derived from two linked surveillance systems providing high ascertainment of both STEC and HUS cases, which provide a more robust estimate. It is likely that there will also be a bias resulting from ascertainment of STEC cases from laboratory specimens, as milder cases of gastrointestinal (GI) infection are less likely to be microbiologically tested, but this will also be true of previously published studies.

While rurality has been reported as an important factor in risk of STEC infection,[2 14] our study suggests that rurality is not a significant driver of progression to HUS. It is important to note that there are environmental factors, such as cattle density, that were not included in this study and which may be more important factors in risk of STEC infection. Our finding that rurality was not linked to progression to HUS following STEC infection may also be due to the majority of our cases (95%) being STEC O157; this finding may be different in more heterogenous datasets from countries with greater variability by serogroup. Similarly, despite evidence to suggest that the risk and consequences of GI infections in general are greater for disadvantaged children,[22–26] the finding in our study suggests that lower childhood SEC is unlikely to be a contributor for development of HUS.

Previous studies in England have suggested that children aged 1–4 years, girls and white ethnic groups have the highest incidence of STEC infection.[2 27] Our study echoes the findings by Milford *et al*,[6] which demonstrated higher progression to HUS among children aged 1–4 years. No overall difference in risk of HUS by sex was identified in our study, a finding echoed in several other previous studies[28–31]; this is an area of disagreement in the literature, with several studies finding higher risk among women,[7 17 31] although two of these studies finding higher risk in women did not look specifically among children.[7 31] We did find differences in risk by sex within specific age groups, with a greater proportion of progression to HUS among girls less than 1 year of age and 10–15 years of age compared with boys of the same age groups (table 1), although no significant interaction between age and sex could be identified. The reasons for the differential risk by age are currently unclear and call for a deeper understanding of differences in risks and exposures between these groups.

The association between clinical presentation with vomiting and bloody diarrhoea and increased risk of HUS reported in this study has been identified previously,[12] and, as such, the presence of these symptoms particularly in paediatric STEC cases should evoke a high level of clinical suspicion for the potential development of HUS.

Our study quantifies the proportion of paediatric STEC cases progressing to HUS in a well-defined population with high ascertainment. It also quantifies the risk factors associated with progression to HUS in terms of sociodemographic characteristics as well as clinical presentation. Further research is warranted to elucidate the populations at risk of STEC infection and HUS in terms of deprivation, ethnicity, age and sex, in order to better understand whether there are real differences in risk or artefacts of surveillance.

**Acknowledgements** The authors thank Kirsten Glen and Naomi Launders for their work on the National Enhanced Surveillance System for Shiga toxin-producing *Escherichia coli*, Ross Harris for statistical advice and Richard Lynn at the British Paediatric Surveillance Unit for facilitating the British Paediatric Surveillance Unit HUS study.

**Contributors** All authors contributed to the conception and design of the study. LB, CJ and BA collated and curated the dataset and provided guidance on the interpretation of data. NA performed the analyses with guidance from AC, LB, BB, JH and DT-R. NA, JH, DTR, MV, SOB and MW drafted the manuscript, which was critically revised by all authors. All authors approved the final version of the manuscript. JH and DTR are joint senior authors.

**Funding** The research was funded by the National Institute for Health Research Health Protection Research Unit in Gastrointestinal Infections at the University of Liverpool in partnership with Public Health England, in collaboration with University of East Anglia, University of Oxford and the Quadram Institute. NA is based at the University of Liverpool and Public Health England.

**Disclaimer** The views expressed are those of the author(s) and not necessarily those of the National Health Service, the National Institute for Health Research, the Department of Health and Social Care or Public Health England.

**Competing interests** None declared.

**Patient consent for publication** Not required.

**Ethics approval** Ethical approval was originally obtained for the main study (ref: 11/LO/1412). As of October 2010, haemolytic uraemic syndrome is a statutory reportable condition, and this study falls under the existing Health Protection Agency (now Public Health England) permissions under Section 251 of the NHS Act 2006. In addition, we received a favourable ethical opinion from the South East Coast—Surrey Research Ethics Committee (15/LO/2138) on 1 December 2015 covering the use of this dataset for this study.

**Provenance and peer review** Not commissioned; externally peer reviewed.

**Data availability statement** No data are available.

**ORCID iD**
Natalie Adams http://orcid.org/0000-0002-6131-480X

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
