## [Reviewer comments · BMJ Paediatrics Open]

This paper was submitted to a another journal from Archives of Disease in Childhood but declined for publication following peer review. The authors addressed the reviewers' comments and submitted the revised paper to BMJ Paediatrics Open. The paper was subsequently accepted for publication at BMJ Paediatrics Open.

ARTICLE DETAILS

TITLE (PROVISIONAL)	Sociodemographic and clinical risk-factors for paediatric typical haemolytic uraemic syndrome
AUTHORS	Adams, Natalie; Byrne, Lisa; Rose, Tanith; Adak, Bob; Jenkins, Claire; Charlett, Andre; Violato, Mara; O'Brien, Sarah; Whitehead, Margaret; Barr, Benjamin; Taylor-Robinson, David; Hawker, Jeremy

VERSION 1 – REVIEW

REVIEWER	Reviewer name: Hoyer, Peter Institution and Country: EAP Competing interests: N/A
REVIEW RETURNED	22-Jan-2019

GENERAL COMMENTS	With regard to HUS, this is indeed a large epidemiological study combining 2 data sets from active surveillance, however clinical data were extracted in a retrospective manner. About 19% of children infected with stx producing E.-coli progressed to HUS. The rate was higher in females and children in the age group 1-4 years. Sociodemographic factors had no influence on the progression. Bloody diarrhea and antibiotic use seem to of predictive value. Therefore, these factors should increase the suspicion for the development of an HUS. The clinical message is important and should merit to be published. I have several questions and concerns: I would be interested to know if socioeconomic factors are a risk factor to acquire stx positive E.-coli enteritis. In my clinical experience, we get more patient with HUS from areas with agriculture where raw milk products are consumed. It is a pity, that the need for dialysis is not reported, and any information on recovery is missed. The readability of the elaborated statistical methods, as well as employed classification, are difficult to follow. I cannot prove them. I am afraid that the majority of ADC readers will have the same problem. In its present form, the manuscript is better addressed and more suitable for epidemiologists or public health experts. If it will be of interest for ADC, I suggest to condense the description of the statistical methods or offer to rewrite it as a letter.
---

VERSION 1 – AUTHOR RESPONSE

REVIEWERS' COMMENTS TO AUTHOR:

With regard to HUS, this is indeed a large epidemiological study combining 2 data sets from active surveillance, however clinical data were extracted in a retrospective manner.

About 19% of children infected with stx producing E.-coli progressed to HUS.

The rate was higher in females and children in the age group 1-4 years.

Sociodemographic factors had no influence on the progression. Bloody diarrhea and antibiotic use seem to of predictive value.

Therefore, these factors should increase the suspicion for the development of an HUS.

The clinical message is important and should merit to be published.

Thank you for your comments.

I have several questions and concerns:

I would be interested to know if socioeconomic factors are a risk factor to acquire stx positive E.-coli enteritis.

Thank you. In our recent paper (Adams et al. 2019) we explored the influence of socioeconomic status on Shiga toxin-producing Escherichia coli infection incidence, risk factors and clinical presentation. We found higher incidence in in the highest socioeconomic status group compared to the lowest. Odds of accident and emergency attendance and hospitalisation were higher in the lowest socioeconomic group. Exposure to foodborne risk factors – salad, fruit, vegetables, UK or non-UK travel and environmental exposures were higher in the highest socioeconomic group compared to the lowest.

In my clinical experience, we get more patient with HUS from areas with agriculture where raw milk products are consumed.

It would have been interesting to explore the relationship between consumption of raw milk and progression to HUS however there was limited data in this dataset with which to explore this. Other studies, such as Elson et al. 2018 have explored the relationship between STEC, rurality and risk factors such as animal density. We added a sentence to the discussion to acknowledge this limitation of our study:

“It is important to note that there are environmental factors, such as cattle density, that were not included in this study and which may be more important factors in risk of STEC infection. Our finding that rurality was not linked to progression to HUS following STEC infection may also be due to the majority of our cases (95%) being STEC O157 – this finding may be different in a more heterogenous dataset from countries with greater variability by serogroup.”

It is a pity, that the need for dialysis is not reported, and any information on recovery is missed.

Thank you. We agree that it would have been interesting to explore this. A prospective cohort study would have allowed for the follow-up of cases to include clinical outcomes and information on recovery however opportunistic data collection through an existing surveillance system was used for this study. Cases entered the cohort on diagnosis of STEC and their HUS status and socio-demographic characteristics were retrospectively linked to their case record.

The readability of the elaborated statistical methods, as well as employed classification, are difficult to follow. I cannot prove them. I am afraid that the majority of ADC readers will have the same problem. In its present form, the manuscript is better addressed and more suitable for epidemiologists or public health experts.

If it will be of interest for ADC, I suggest to condense the description of the statistical methods or offer to rewrite it as a letter.

Thank you for the comments. As the reviewer suggests this is a relatively complex epidemiological analysis and condensing the methods is not possible without losing transparency and is more suitable for an epidemiological audience, as such we will seek to submit the manuscript elsewhere.

VERSION 2 – REVIEW

REVIEWER	Reviewer name: Jeremy Miles Institution and Country: Google, USA Competing interests: No competing interests
REVIEW RETURNED	18-Feb-2019

GENERAL COMMENTS	This is a clear and well written paper. I have two minor comments. 1. The authors mention the missing at random assumption. But it is assumed that the data are either missing at random, or missing completely at random (MCAR). It is assumed that the data are not missing not at random (mnar). Testing for MNAR is impossible, and MNAR is statistically indistinguishable from MCAR. MCAR mean that no variables predict missingness. MNAR means that variables that are not present in the data predict missingness. 2. Using Poisson regression with binary outcomes is often preferred, because of the improvement of interpretability. My favorite example: if the probability of progression is 0.4 in one group, and 0.6 in the second group, the relative risk (given by Poisson regression) is 1.5. The odds ratio (as given by logistic regression) is 2.33. 1.5 makes more sense. In Stata, one can use the , robust option. See: Zou G. A Modified Poisson Regression Approach to Prospective Studies with Binary Data. Am J Epidemiol 2004; 159(7):702-6.
---

REVIEWER	Reviewer name: Mathias Bruyand Institution and Country: Santé publique France Competing interests: None
REVIEW RETURNED	01-Apr-2019

GENERAL COMMENTS	Main comments This manuscript aims at investigating the relationship between demographic factors and pediatric HUS occurrence. 207 HUS cases were included in the study, giving a sufficient statistical power to perform the analysis. All the results provided regarding significant associations were already described and are not originals, the fact that the main variables of interest were not related to HUS risk in adjusted analysis could be due to the fact that some major confusion factors (STEC serogroup) were not taken into account. Specific comments Introduction I agree that STEC encoding stx2 are more frequently associated with HUS than other strains. However, the STEC serogroup is a major predictor of HUS, as the STEC O157 serogroup has been early identified as highly virulent, and as some serogroups are not associated with HUS (please see for example Karmali MA et al. Association of genomic O island 122 of Escherichia coli EDL 933 with verocytotoxin-producing Escherichia coli seropathotypes that are linked to epidemic and/or serious disease. J Clin Microbiol. 2003 Nov;41(11):4930-40.). I believe that if the authors are describing STEC characteristics, the serogroup should be described as well. In this section, the rationale of the objective is not developed: why do the authors believe that demographic factors should be involved in HUS occurrence? Which are the hypotheses? Which characteristics may be involved? Methods section Study design A retrospective cohort was set by linking two data sources. A longitudinal follow-up is expected with this design and I do not understand why the risk of progression to HUS was not assessed through Poisson regression or survival analysis. Or perhaps this was not a cohort study. Another point is unclear: which information was gathered for which database? There is some overlap between information gathered in NESS and BPSU: both collect HUS diagnosis and demographics for example. If the HUS diagnosis is collected in the NESS database, why was the BPSU database used? What happened when there were discordances between the two databases regarding HUS diagnoses (HUS in one database and not in the other)? Case definition: the case definition of the outcome (HUS) should be detailed in the manuscript. Are the HUS cases definitions used in the two databases the same? Was a specific diagnosis ascertainment procedure used for this study? Confounding factors: ruminants account for the main reservoir for some virulent STEC serogroups (O157, O26), a differential exposure is expected between children living in rural areas and in urban areas. These serogroups are more frequently complicated by HUS than other serogroups. Thus, serogroup may act as an important confusion factor in this analysis (associated with HUS and with rurality), analysis should be adjusted on it. Robustness analysis: the authors precise in the robustness tests paragraphs of the methods section that the relationship between age and sex in the cohort was assessed through a fractional polynomial prediction to detect the best functional form for age as a continuous variable. We see the results of this analysis in the supplementary fig 1. However, the results of this analysis are not cited in the results section, and age is not used as a continuous variable, but as a categorical variable. This is very unclear.
--

	Results section The proportion of HUS is higher than expected (nearly 20%) while 15% is expected. Descriptive analysis, last sentence: these results should be developed ($p=0.07$ is almost significant), the results with CI should be cited so we can see the trends if any. Adjusted analyses: The authors say that there was no difference in risk by rurality or by region, results should be cited. Discussion The authors indicate that they may have an improved case ascertainment than in prior studies, but case ascertainment was not cited before the discussion. This point (case ascertainment) should be developed before. The proportion of HUS was higher than expected and this could be the result of misclassifications due to problems in case ascertainment. Was a sample of medical files checked to assess the validity of the HUS diagnoses recorded? The authors state p10 57 that the results are likely to be generalizable to other countries, p11 31, the authors say : in England, most diagnosed cases of STEC are O157, and may therefore not be directly applicable to countries where other serogroup predominate. I believe that there is a contradiction between these two sentences. Here again, serogroup should be taken into account in the analyses. This can act as a confusion factor and explain the fact that associations significant in univariable analyses (IMD) were no longer significant in adjusted analysis. Furthermore, the role of all socioeconomic factors has not been explored, but in many countries there is a heterogeneity of pediatric HUS incidence between regions, and a link between pediatric HUS incidence and cattle density has been shown. Thus in this study, differences in HUS risk by rurality or by region may be expected. Here again, a confusion bias could explain the fact that these variables were not associated with HUS risk in adjusted analyses.
--	--

REVIEWER	Reviewer name: Pia Hardelid Institution and Country: UCL Great Ormond Street Institute of Child Health, UK Competing interests: None
REVIEW RETURNED	24-Apr-2019

GENERAL COMMENTS	This is a paper where linked data from two national surveillance databases are analysed to examine risk factors for progression to haemolytic uraemic syndrome in children with STEC. I have a few comments which will hopefully improve the manuscript and the messages. The biggest challenge is the focus on ethnic group when the grouping is so crude. Introduction 1) It would be helpful for to explain how STEC is spread (this could be done very briefly) – this would help explain why your hypotheses are that poorer children, children aged 1-4 and children living in rural areas would be at higher risk of severe infection Methods 2) The linked data sources are unique and more could be made of this. There is no mention of what % of children in either dataset were linked. Could you provide more information on this? Why did you only link on NHS number? Has this been shown to be complete and accurate in both data sources? Please provide more info about the linkage
--

	3) On a related point, the authors should use a flow chart to explain how the final study size was arrived at, that is, how many children were on the NESSS, how many children in HUS, and then show the various exclusions. 4) You are not checking whether data are MAR (this is not possible, see for example Potthoff et al, 2016. Statistical Methods in Medical Research, 15, 2013-234), you are checking whether data are Missing Completely At Random (MCAR). MCAR is not a necessary assumption for MI. Why are you showing these results? Instead, can you explain why you assume your data are MAR, rather than missing not at random? 5) Your sample size is not huge so I don't understand why you are testing for effect modification (or, indeed, split your Table 1 by age AND sex). If you are going to test for effect modification between age and sex you would need to explain very clearly why you are hypothesising that these would exist? Otherwise, I suggest strongly not testing for effect modification and putting less emphasis on the differences between sex and age – sex is not associated with HUS in any case. 6) White and non-white ethnicity is extremely crude and would hide large differences in terms socio-economic status and migration history among both groups. You need to explain why you have chosen this categorisation – is this due to numbers or data quality for the ethnicity variable? I would tone down the ethnic group analyses and discussion of findings if you are going to use this very crude binary variable – it is simply not helpful for understanding the epidemiology. Results 7) What statistical test do the p-values refer to? 8) There are an awful lot of sensitivity analyses and I am not entirely clear what the point of them all are (eg why do an analysis excluding ethnic group? Either it needs to be adjusted for or not, surely? But see point 6 above) Discussion 9) Please cite studies discussing the usefulness of IMD as an indicator of SES. You say it is preferable as it measures area-level effects of deprivation, but whether that is useful or not depends on your hypotheses regarding what is driving potential inequalities in outcomes. Is it area-level effects or individual level effects (eg poor housing, exposure to certain types of high-risk foods etc) This is why you should state your hypothesis clearly in the intro 10) Surely one weakness is that you don't know whether the children had underlying chronic conditions or not – presumably this would be a major predictor of progression to severe disease
--	---

VERSION 2 – AUTHOR RESPONSE

Reviewer: 1

This is a clear and well written paper. I have two minor comments.

1. The authors mention the missing at random assumption. But it is assumed that the data are either missing at random, or missing completely at random (MCAR). It is assumed that the data are not missing not at random (mnar). Testing for MNAR is impossible, and MNAR is statistically indistinguishable from MCAR. MCAR mean that no variables predict missingness. MNAR means that variables that are not present in the data predict missingness.

Thank you. In terms of the missingness mechanism, we believe that the data are either missing at random or missing completely at random, and like Pedersen et al think that MAR is more plausible (<https://www.ncbi.nlm.nih.gov/pmc/articles/PMC5358992/>). Multiple imputation using chained equations (MICE) operates under the assumption that, given the variables used in the imputation procedure, the missing data are missing at random, i.e. that the probability that a value is missing depends only on observed values and not on unobserved values (Schafer & Graham, 2002). However, MICE provides unbiased estimation when data are MCAR (Pedersen et al). We believe that the for the variables in our study missingness can be accounted for by variables where there is information. Whilst missing at random is an assumption that cannot be definitively tested for statistically, assessing the distribution of missing and not missing ethnicity by age, sex and region is useful to understand if a mechanism other than MCAR or MAR is indicated.

Moreover, due to the proportion of missing data for ethnicity and the use of multiple imputation to account for this, a sensitivity analysis was undertaken excluding the ethnicity variable to explore whether the inclusion of the multiply imputed ethnicity variable modified the relationship between SEC and development of HUS (Supplementary Table 3). While the missing mechanism may be MCAR for certain variables and MAR for others, MICE provides unbiased estimates for either or a mixture.

2. Using Poisson regression with binary outcomes is often preferred, because of the improvement of interpretability. My favorite example: if the probability of progression is 0.4 in one group, and 0.6 in the second group, the relative risk (given by Poisson regression) is 1.5. The odds ratio (as given by logistic regression) is 2.33. 1.5 makes more sense.

In Stata, one can use the `poisson, robust` option. See: Zou G. A Modified Poisson Regression Approach to Prospective Studies with Binary Data. *Am J Epidemiol* 2004; 159(7):702-6.

Thank you. While you are correct that risk ratio estimates can be obtained using robust quasi-Poisson models, they can also be directly estimated from a binomial model with a log link, although problems with convergence are frequently encountered. We preferred to assess associations using a logistic regression model for our study. As the outcome (development of HUS) is rare the odds ratio provides a reasonable approximation of the risk ratio so if they were to be mis-interpreted as risk ratios by readers this wouldn't be too inappropriate. Additionally, the mapping of the risk to the whole of the real line that occurs with the logit link often overcomes issues of scale dependent effect modification and protection against differential reporting biases.

Reviewer: 2

Main comments

This manuscript aims at investigating the relationship between demographic factors and pediatric HUS occurrence. 207 HUS cases were included in the study, giving a sufficient statistical power to perform the analysis. All the results provided regarding significant associations were already described and are not originals, the fact that the main variables of interest were not related to HUS risk in adjusted analysis could be due to the fact that some major confusion factors (STEC serogroup) were not taken into account.

Thank you. We did not explore the relationship with serogroup specifically for several reasons. In England, during the time-frame of this study, culture methods were predominantly used at frontline diagnostic laboratories to specifically detect STEC O157 strains and other serogroups were not routinely tested for, except in cases of HUS where faecal specimens were sent to the national reference laboratory for detection of all serogroups. Consequently, 95% of STEC cases detected were STEC O157.

Only 46 cases of STEC where a serogroup other than O157 was detected were included, and these comprised twelve cases for which the STEC serogroup was unidentifiable, with the remaining cases being comprised of 12 different serogroups.

Due to the low recovery of non-O157 serogroups, case numbers are too low to delineate by serogroup and would be misleading as non-O157 is more often recovered from HUS cases than non HUS cases. Analysing the data as a group of O157 versus non-O157 strains would also be inappropriate as they are heterogenous. We did however analyse by Shiga-toxin type, (Stx), which the latest evidence shows is a major organism-related virulence factor in the development of HUS and, as a key predictor of O type virulence, can give some information that can be extrapolated to) types that were not well-represented in the English dataset used by us.

Specific comments

Introduction

I agree that STEC encoding *stx2* are more frequently associated with HUS than other strains. However, the STEC serogroup is a major predictor of HUS, as the STEC O157 serogroup has been early identified as highly virulent, and as some serogroups are not associated with HUS (please see for example Karmali MA et al. Association of genomic O island 122 of *Escherichia coli* EDL 933 with verocytotoxin-producing *Escherichia coli* seropathotypes that are linked to epidemic and/or serious disease. *J Clin Microbiol.* 2003 Nov;41(11):4930-40.). I believe that if the authors are describing STEC characteristics, the serogroup should be described as well.

We agree that serogroup is of interest, but as 95% of our cases were O157 (seropathotype A, which was reported by Karmali as 100% positive for OI-122) and the remaining 5% were heterogeneous (or unidentifiable) by serotype, we are unable to do this using our English dataset. We have added a line to the discussion as follows:

“In England, most diagnosed cases of STEC are of serogroup O157 (95% in our study), and it is possible that our results may be biased towards the relationship between STEC O157 and progression to HUS, which may differ if other, possibly less pathogenic, serogroups predominate.”

In this section, the rationale of the objective is not developed: why do the authors believe that demographic factors should be involved in HUS occurrence? Which are the hypotheses? Which characteristics may be involved?

Thank you, we have revised this section to make the rationale of the study and its hypotheses clearer.

“...however few have documented progression to HUS by other demographic characteristics such as deprivation, foreign travel, rurality or region. There is evidence to suggest that those who are disadvantaged have a lower risk of STEC infection (Chang et al., 2009, Jalava et al., 2011, Whitney et al., 2015), and potentially a lower risk of progression to HUS outside of England (Rowe et al., 1991, Whitney et al., 2015), however no studies have looked at the relationship between SES, STEC and HUS in England.”

Methods section

Study design

A retrospective cohort was set by linking two data sources. A longitudinal follow-up is expected with this design and I do not understand why the risk of progression to HUS was not assessed through Poisson regression or survival analysis. Or perhaps this was not a cohort study.

Thanks. Cases entered the cohort on diagnosis of STEC and their HUS status and socio-demographic characteristics were retrospectively linked to their case record. Due to the nature of the two surveillance systems, it is not possible to prospectively follow up STEC cases as HUS data is routinely collected some time after the STEC data. While STEC cases were not followed up to see if they subsequently developed HUS, any STEC case that developed HUS should be picked up by one or both data sources. We have added a new figure, Supplementary Figure 1, which describes the flow of participants in this study and which we believe improves the clarity of the description of the study design. As described in response to the comment above by Reviewer 1 regarding the statistical analysis of this cohort, we preferred to assess associations using a logistic regression model for our study. We accept there may be advantages to using alternative models in cohort studies. In our study we are not specifically interested in time to HUS but whether this sequelae happened. As there is no censoring of outcome we feel there is little additional benefit to be gained for using proportional hazard models over that of a logistic model.

Another point is unclear: which information was gathered for which database? There is some overlap between information gathered in NESS and BPSU: both collect HUS diagnosis and demographics for example. If the HUS diagnosis is collected in the NESS database, why was the BPSU database used? What happened when there were discordances between the two databases regarding HUS diagnoses (HUS in one database and not in the other)?

Thank you. We agree that this was unclear and have added additional detail into the methods to improve clarity of this point, as follows:

“The linkage of two robust datasets, both of which can record HUS status, ensures high ascertainment of HUS cases.”

“Due to the timing of the ESQ administration in NESS (which is designed to inform the acute public health response), this system can under-ascertain HUS as this can develop after completion of the questionnaire.”

We also believe that the addition of Supplementary Figure 1, described above, assists with the interpretation of the linkage.

Case definition: the case definition of the outcome (HUS) should be detailed in the manuscript. Are the HUS cases definitions used in the two databases the same? Was a specific diagnosis ascertainment procedure used for this study?

The clinical case definition used in the BPSU study is included in Supplementary Tables 1a and 1b.

Confounding factors: ruminants account for the main reservoir for some virulent STEC serogroups (O157, O26), a differential exposure is expected between children living in rural areas and in urban areas. These serogroups are more frequently complicated by HUS than other serogroups. Thus, serogroup may act as an important confusion factor in this analysis (associated with HUS and with rurality), analysis should be adjusted on it.

We agree that serogroup is of interest, but as 95% of our cases were O157 and the remaining 5% were heterogeneous (or unidentifiable) by serogroup, we are unable to do this using our English dataset. As described above in response to an earlier comment, we have added a line in the discussion to describe this.

Robustness analysis: the authors precise in the robustness tests paragraphs of the methods section that the relationship between age and sex in the cohort was assessed through a fractional polynomial prediction to detect the best functional form for age as a continuous variable. We see the results of this analysis in the supplementary fig 1.

However, the results of this analysis are not cited in the results section, and age is not used as a continuous variable, but as a categorical variable. This is very unclear.

Thank you. In response to this and comments by reviewer 3, we have reduced the number of sensitivity analyses presented in order to improve the clarity of the manuscript as we agree with reviewer 3 that these analyses do not add to the findings of this study

Results section

The proportion of HUS is higher than expected (nearly 20%) while 15% is expected.

Descriptive analysis, last sentence: these results should be developed ($p=0.07$ is almost significant), the results with CI should be cited so we can see the trends if any.

Thank you. We have now amended the descriptive results section to include the confidence intervals for the proportion progressing to HUS in all 5 quintiles as follows:

“Although progression to HUS was higher in the least disadvantaged quintile (47/245, 19.2%, 95% CI 14.4-24.7%) compared with the most disadvantaged quintile (29/189, 15.3%, 95% CI 10.5-21.3%) this difference was not statistically significant ($p=0.29$). The highest proportion progressing to HUS was in quintile 3 (53/219, 24.2%, 95% CI 18.7-30.4%) and there was no clear pattern across the 5 quintiles ($p=0.07$; quintile 2 - 35/221, 15.8%, 95% CI 11.3-21.3%; quintile 4 - 43/185, 23.2%, 95% CI 17.4-30%).”

Adjusted analyses: The authors say that there was no difference in risk by rurality or by region, results should be cited.

Thank you. We have added the results for rurality into the text and have referenced Table 2 for results by region due to the number of regions included in the analysis.

Discussion

The authors indicate that they may have an improved case ascertainment than in prior studies, but case ascertainment was not cited before the discussion. This point (case ascertainment) should be developed before. The proportion of HUS was higher than expected and this could be the result of misclassifications due to problems in case ascertainment. Was a sample of medical files checked to assess the validity of the HUS diagnoses recorded?

Thank you. We have added a sentence to the methods to explain that the linkage of the two systems ensures high case ascertainment as both systems can record HUS. The enhanced surveillance system, NESSS, may under-ascertain HUS due to the timing of the administration of the enhanced surveillance questionnaire as HUS may develop after this point as it is primarily designed for the surveillance of STEC, not HUS. The BPSU study is an active surveillance system for HUS which, by its design, has a high case-ascertainment but cases may lack socio-demographic data.

“The linkage of two robust datasets, both of which can record HUS status, ensures high ascertainment of HUS cases....Due to the timing of the ESQ administration in NESSS, this system can under-ascertain HUS as this can develop after completion of the questionnaire.”

HUS cases were clinically diagnosed using robust clinical case definitions (Supplementary Table 1a and 1b).

We have acknowledged in the discussion that there is likely to be a bias in the ascertainment of STEC cases from laboratory specimens as milder cases of gastroenteritis are less likely to be tested.

The authors state p10 | 57 that the results are likely to be generalizable to other countries, p11 | 31, the authors say : in England, most diagnosed cases of STEC are O157, and may therefore not be directly applicable to countries where other serogroup predominate. I believe that there is a contradiction between these two sentences.

Thank you. We believe the results are generalisable however it is possible that, in countries with other predominant serotypes, the results may differ. We agree that these statements are confusing and have rephrased the argument as follows:

“In England, most diagnosed cases of STEC are of serogroup O157 (95% in our study), and it is possible that our results may be biased towards the relationship between STEC O157 and progression to HUS, which may differ if other, possibly less pathogenic, serogroups predominate.”

Here again, serogroup should be taken into account in the analyses. This can act as a confusion factor and explain the fact that associations significant in univariable analyses (IMD) were no longer significant in adjusted analysis.

As described above in response to an earlier comment, it was not possible to undertake any analysis based on serogroup due to the predominance of O157 (95%).

Furthermore, the role of all socioeconomic factors has not been explored, but in many countries there is an heterogeneity of pediatric HUS incidence between regions, and a link between pediatric HUS incidence and cattle density has been shown. Thus in this study, differences in HUS risk by rurality or by region may be expected. Here again, a confusion bias could explain the fact that these variables were not associated with HUS risk in adjusted analyses.

Thank you for your comment. We have added a sentence to the discussion to acknowledge this as follows:

“It is important to note that there are environmental factors, such as cattle density, that were not included in this study and which may be more important factors in risk of STEC infection. Our finding that rurality was not linked to progression to HUS following STEC infection may also be due to the majority of our cases (95%) being STEC O157 – this finding may be different in more heterogeneous dataset from countries with greater variability by serogroup.”

Reviewer: 3

This is a paper where linked data from two national surveillance databases are analysed to examine risk factors for progression to haemolytic uraemic syndrome in children with STEC. I have a few comments which will hopefully improve the manuscript and the messages. The biggest challenge is the focus on ethnic group when the grouping is so crude.

Thank you for your comments, which we have addressed in full underneath each point below, including specifically regarding the crude ethnicity grouping.

Introduction

1) It would be helpful for to explain how STEC is spread (this could be done very briefly) – this would help explain why your hypotheses are that poorer children, children aged 1-4 and children living in rural areas would be at higher risk of severe infection

Thank you. We have added a sentence to the introduction to describe the transmission pathways:

“Transmission to humans occurs through consumption of contaminated food or water, exposure to a contaminated environment involving direct or indirect contact with animals or their faeces and person-to-person spread.”

Methods

2) The linked data sources are unique and more could be made of this. There is no mention of what % of children in either dataset were linked. Could you provide more information on this? Why did you only link on NHS number? Has this been shown to be complete and accurate in both data sources? Please provide more info about the linkage

Thank you. We agree that this information would improve the clarity of the manuscript and, in response to the comment below, we have added a flow chart describing the process of linking the two datasets, including the numbers of children at each stage of the process. We have also clarified in the manuscript that NHS number was used as it was available for all cases.

3) On a related point, the authors should use a flow chart to explain how the final study size was arrived at, that is, how many children were on the NESSS, how many children in HUS, and then show the various exclusions.

Thank you. We agree that this would greatly improve the clarity of the methods and we have now added a new figure ‘Supplementary Figure 1’ to the submission which describes the selection of participants into the study.

4) You are not checking whether data are MAR (this is not possible, see for example Potthoff et al, 2016. *Statistical Methods in Medical Research*, 15, 2013-234), you are checking whether data are Missing Completely At Random (MCAR). MCAR is not a necessary assumption for MI. Why are you showing these results? Instead, can you explain why you assume your data are MAR, rather than missing not at random?

Thank you - Please see the detailed response on the subject of missingness to the comments made by reviewer 1.

5) Your sample size is not huge so I don’t understand why you are testing for effect modification (or, indeed, split your Table 1 by age AND sex). If you are going to test for effect modification between age and sex you would need to explain very clearly why you are hypothesising that these would exist? Otherwise, I suggest strongly not testing for effect modification and putting less emphasis on the differences between sex and age – sex is not associated with HUS in any case.

Thank you, your points regarding statistical power and biologic plausibility for effect modification is well made and we have reduced these analyses in the paper.

6) White and non-white ethnicity is extremely crude and would hide large differences in terms socio-economic status and migration history among both groups. You need to explain why you have chosen this categorisation – is this due to numbers or data quality for the ethnicity variable? I would tone down the ethnic group analyses and discussion of findings if you are going to use this very crude binary variable – it is simply not helpful for understanding the epidemiology.

Unfortunately ethnicity is not well-recorded in either of the datasets used in this study. We have added a sentence to the discussion to reflect this as follows:

“There were also some missing data in our study, particularly for ethnicity, which we addressed using multiple imputation. The ethnicity variable used (White/Non-White) was also crude and adopted because of data quality issues. This may mask differences in socioeconomic status.”

In response to the previous comment, we have also reduced the number of sensitivity analyses and as such have reduced the emphasis on exploring the relationship between HUS and ethnicity.

Results

7) What statistical test do the p-values refer to?

We used the chi-squared test to compare the proportions in descriptive analysis. We have added this to the methods. We have also added a footnote to the p-value column in Table 2 to clarify that the statistical significance of the relationship between HUS and each variable was tested using the χ^2 test.

8) There are an awful lot of sensitivity analyses and I am not entirely clear what the point of them all are (eg why do an analysis excluding ethnic group? Either it needs to be adjusted for or not, surely? But see point 6 above)

Thank you – as per the responses above, we have reduced the number of sensitivity analyses to explore only the effect of excluding travel associated cases and ethnicity.

Discussion

9) Please cite studies discussing the usefulness of IMD as an indicator of SES. You say it is preferable as it measures area-level effects of deprivation, but whether that is useful or not depends on your hypotheses regarding what is driving potential inequalities in outcomes. Is it area-level effects or individual level effects (eg poor housing, exposure to certain types of high-risk foods etc) This is why you should state your hypothesis clearly in the intro

Thank you. In response to your previous comment, detailed above, we have added more detail into the introduction to clarify our hypotheses. We have also, in response to your earlier comment regarding transmission routes for STEC, added some background to this into the introduction which demonstrates that various area-level factors such as environmental exposures to animals or their faeces are important transmission pathways. Furthermore, no individual-level measures of deprivation were available in either dataset.

We have also expanded on this section in the discussion as follows:

“Further, as an area-level measure of SEC was used, it is possible that it may not have been sensitive enough to detect the effect of socioeconomic inequalities, particularly if individual factors rather than area-level factors have more influence over the risk of acquiring more severe strains of STEC with increased risk of progression to HUS. However, person-to-person spread is an important risk factor for GI infections and, although there is a risk of ecological fallacy, area-level measures have the advantage of including potential environmental factors such as housing and living environment deprivation which are likely to be important factors in considering individual risk of infection.”

10) Surely one weakness is that you don't know whether the children had underlying chronic conditions or not – presumably this would be a major predictor of progression to severe disease

Thank you, we have now included this in the discussion as a limitation of the study:

“No data were available on whether the children included in our study had underlying or chronic conditions which may be related to their risk of developing HUS.”

VERSION 3 – REVIEW

REVIEWER	Reviewer name: Mathias Bruyand Institution and Country: Santé publique France, France Competing interests: None
REVIEW RETURNED	25-Jun-2019

GENERAL COMMENTS	I thank the authors, all my comments were correctly addressed.
--

REVIEWER	Reviewer name: Pia Hardelid Institution and Country: UCL Great Ormond Street Institute of Child Health Competing interests: None
REVIEW RETURNED	08-Jul-2019

GENERAL COMMENTS	Thank you for your comments and for taking the suggestions of the reviewers on board - i hope you feel it has improved the manuscript. I am still not entirely clear why you adjust for a very crude ethnic group variable, and what you are hoping to explain in terms of variability in outcomes using such a crude indicator, particularly as you say, when there are small numbers and missing data for this variable. If you are still planning to include ethnic group in your model, please could you include a justification in your methods for what potential difference in risk you are hoping to explain using such a crude indicator. Are there particular exposures that you think will be different in these two very large and internally diverse groups that this crude binary variable will capture? If so, what are they? The following paper may be of interest: https://www.ncbi.nlm.nih.gov/pubmed/24887159
--

VERSION 3 – AUTHOR RESPONSE

Reviewer: 2

Thank you for your comments and for taking the suggestions of the reviewers on board - i hope you feel it has improved the manuscript.

I am still not entirely clear why you adjust for a very crude ethnic group variable, and what you are hoping to explain in terms of variability in outcomes using such a crude indicator, particularly as you say, when there are small numbers and missing data for this variable. If you are still planning to include ethnic group in your model, please could you include a justification in your methods for what potential difference in risk you are hoping to explain using such a crude indicator. Are there particular exposures that you think will be different in these two very large and internally diverse groups that this crude binary variable will capture? If so, what are they? The following paper may be of interest:
<https://www.ncbi.nlm.nih.gov/pubmed/24887159>

Thanks. Previous analyses have demonstrated differences in STEC between white ethnic groups and non-white ethnic groups, so we used this indicator in our study. For example, in Byrne et al 2015 (RR 1.43, P < 0.001). We have added a sentence to the paper as suggested to make this clearer in the methods:

P7: "Ethnic groups, collected in five categories (White, Asian/Asian British, Black/Black British, Mixed, Chinese) is not well-completed in NESSS and therefore responses were re-coded as White or non-White for analysis. The considerable missing data for the ethnicity variable (19.1%) has led us to use the crude dichotomy of White/non-White in this analysis. Multiple imputation using chained equations was used to impute values where ethnicity (White/non-White) was missing. There will clearly be some loss of information from doing this, and this precludes investigating risk differences between the non-White ethnic groups. This may also slightly affect the confounding that exists between ethnicity and socioeconomic status."

And we have also added a sentence in the limitations outlining the limitations of the ethnicity data:

P12: "The binary ethnicity variable used (White/non-White) was also crude and adopted because of data quality issues in NESSS for this variable. However, a previous study using this data (2) demonstrated differences in risk of STEC between White and non-White ethnic groups (RR 1.43, $p < 0.001$) and so was important to assess in our study although its inclusion may mask differences in socioeconomic status."